# A Flexible Wearable Strain Sensor Based on Nano-Silver-Modified Laser-Induced Graphene for Monitoring Hand Movements

**DOI:** 10.3390/mi15080989

**Published:** 2024-07-31

**Authors:** Mian Zhong, Yao Zou, Hongyun Fan, Shichen Li, Yilin Zhao, Bin Li, Bo Li, Yong Jiang, Xiaoqing Xing, Jiaqing Shen, Chao Zhou

**Affiliations:** 1Institute of Electronic and Electrical Engineering, Civil Aviation Flight University of China, Deyang 618307, Chinaxingxiaoqing@cafuc.edu.cn (X.X.); 13881017769@163.com (J.S.); 2Faculty of Chemical Engineering, Kunming University of Science and Technology, Kunming 650500, China; zhaoyl@stu.kust.edu.cn (Y.Z.);; 3School of Physics, University of Electronic Science and Technology of China, Chengdu 611731, China; 4School of Mathematics and Physics, Southwest University of Science and Technology, Mianyang 621010, China; y_jiang@swust.edu.cn

**Keywords:** LIG, doping modification, silver nanoparticles, flexible wearable strain sensor, single step in situ

## Abstract

The advancement in performance in the domain of flexible wearable strain sensors has become increasingly significant due to extensive research on laser-induced graphene (LIG). An innovative doping modification technique is required owing to the limited progress achieved by adjusting the laser parameters to enhance the LIG’s performance. By pre-treating with AgNO_3_, we successfully manufactured LIG with a uniform dispersion of silver nanoparticles across its surface. The experimental results for the flexible strain sensor exhibit exceptional characteristics, including low resistance (183.4 Ω), high sensitivity (426.8), a response time of approximately 150 ms, and a relaxation time of about 200 ms. Moreover, this sensor demonstrates excellent stability under various tensile strains and remarkable repeatability during cyclic tests lasting up to 8000 s. Additionally, this technique yields favorable results in finger bending and hand back stretching experiments, holding significant reference value for preserving the inherent characteristics of LIG preparation in a single-step and in situ manner.

## 1. Introduction

With the advancement of science and technology, the limitations of traditional wearable devices are becoming increasingly apparent. These limitations include their bulky size, poor data accuracy, and inability to meet users’ demands for portability and precise monitoring. In this context, flexible wearable devices have emerged as a compact, simple solution capable of providing more accurate and diverse physiological monitoring functions [1]. Current research on flexible wearable strain sensors primarily focuses on designing innovative materials such as MXene [2], conductive hydrogels [3], and graphene [4], which are used to prepare highly sensitive sensors for detecting human physiological signals and movements. Among these materials, graphene [5] stands out due to its exceptional optical, electrical, and mechanical properties, which make it a revolutionary material for future applications [6].

Graphene preparation methods include chemical gas deposition [7], liquid phase separation [8], and epitaxial growth [9]. However, these methods often involve complexity and high costs. The method of laser-induced graphene (LIG) is notable for its simplicity and low cost [10]. LIG exhibits excellent conductivity, mechanical flexibility, and environmental stability. It possesses a three-dimensional porous structure that makes it ideal for flexible wearable strain sensors. The properties of LIG are influenced by various factors such as the type of carbon precursor [11,12,13], laser parameters [14,15,16], and doping modifications [17,18], which can significantly impact performance. Thus, it is crucial to study these influencing factors and select the appropriate materials and parameters to achieve high-performance LIG.

The main research directions involve systematic investigations of laser parameters and doping modification experiments on LIG. By exploring the effect of the laser on the carbon precursor surface, we can understand the evolution of the LIG’s microstructure [19]. Additionally, controlling the laser parameters enables the regulation of the LIG’s microstructure and surface characteristics during preparation, leading to optimized performance [20]. However, studying the LIG’s performance remains a challenge. Doping modification provides a means to enhance or improve specific properties of the LIG, thereby expanding its potential applications in flexible wearable strain sensors. Therefore, numerous research teams are currently focusing on enhancing the performance of LIG-based flexible sensors through doping modifications. For instance, the Robert’s team [21] utilized chemical deposition techniques to fabricate LIG integrated with platinum nanoparticles, enabling the development of lactate amperometry and coulombic potassium test-strip sensors. These sensors exhibit selective detection capabilities for saliva samples collected within the physiological range. Nguyen’s team [22] developed an ion-selective electrode comprising porous LIG and molybdenum disulfide, which incorporates a sensitive ion–electron transducer for measuring potassium ion concentration in hydroponic nutrient solutions within greenhouse environments. Chen’s team [23] proposed a formaldehyde electrochemical sensor consisting of LIG-based detector electrodes modified with nano-silver using a metal ion exchange method. The current research primarily focuses on introducing target objects such as Pt, MoS_2_, and carbon nanotubes to enhance specific properties. However, this approach increases the experimental complexity and can introduce instability and inconsistency into the preparation process. Therefore, optimizing doping technology can achieve significant performance improvements and promote the widespread application of LIG in flexible wearable strain sensors.

Based on our previous study [24,25], we propose the concept of introducing nano-Ag particles with excellent conductivity into LIG through a single-step preparation process. By pre-treating the carbon precursor polyimide (PI) to incorporate Ag ions onto its surface, nanometer-sized silver particles are doped into the generated LIG via laser-induced experiments. Subsequently, we investigate the effects of doped Ag+ on the LIG’s characteristics, performance, and application in flexible wearable sensors. The main innovations and contributions of this study can be summarized as follows:We propose a nano-Ag-modified LIG flexible wearable strain sensor, which offers a novel approach for the in situ, single-step preparation of LIG and the fabrication of high-performance flexible sensors.The nano-Ag-modified LIG flexible strain sensor exhibits exceptional characteristics, including low resistance, superior sensitivity, excellent stability, and remarkable repeatability.The high-performance flexible wearable strain sensor can accurately characterize finger-bend angles. Additionally, based on its exceptional electric heating performance, it can be further expanded to the field of hand heating and insulation for pilots in cold environments, providing a more comfortable and safe flight experience for pilots.

## 2. Materials and Methods

### 2.1. Preparation of LIG

The experimental process for the preparation of doped silver nanoparticles with uniform distribution in LIG is illustrated in Figure 1. Initially, a 125 μm thick polyimide (PI) film (Shenzhen Jihongda Plastic Products Co., Ltd., Shenzhen, China) was immersed in deionized water and subjected to ultrasonic cleaning for 10 min using an ultrasonic cleaner (Chun Rain Inc., Shenzhen, China). The PI film was then dried at a constant temperature of 50 °C for 30 min in a drying oven at constant heat (Shaoxing Subo Instrument Co., Ltd., Shaoxing, China). Subsequently, sodium hydroxide (NaOH) was dissolved in ionized water within a beaker and mixed thoroughly using a magnetic stirrer to prepare a NaOH solution with a concentration of 2.5 mol/L. The PI film was then immersed in the NaOH solution, which led to the breakdown of imide groups on its surface and the formation of polyamide acid along with sodium salt. Subsequently, the PI film was immersed in AgNO_3_ solutions with concentrations of 0.01, 0.03, and 0.05 mol/L, respectively, each for 30 min, to replace Na ions from the sodium salts with Ag ions from the AgNO_3_ solution. After each immersion, the film was cleaned and dried with ionized water. The pre-processed PI was fixed on a stainless steel plate using polyethylene terephthalate (PET) and clamped onto a three-dimensional(3D) moving axis. A CO_2_ infrared laser (Synrad P150, Novanta Corporation, Bedford, MA, USA) was employed to directly irradiate the surface of the PI through a reflective mirror and a vibrating mirror. The irradiated area underwent photothermal action, resulting in chemical reactions. Among these reactions, the CO_2_ infrared laser with specific parameters including a wavelength of 10.64 μm, defocus of 495 mm, laser power of 12.4 W, scanning speed of 105 mm/s, laser repetition rate of 20 kHz, and one scanning cycle, played a crucial role. It is important to note that temperatures exceeding 300 °C during the salt preparation process may lead to the decomposition of silver ions, resulting in the formation of individual silver atoms. However, for LIG production and generation through laser experiments, a significantly higher temperature threshold (>300 °C) is required. This facilitates the breakdown of individual silver atoms while doping LIG with nanometer-sized silver particles.

### 2.2. Characterization of LIG

In this study, the microscopic surface features of LIG were investigated using a Thermo Scientific Helios 5 CX scanning electron microscope (SEM, Thermo Fisher Scientific Inc., Waltham, MA, USA). Energy dispersive X-ray spectroscopy (EDS, Thermo Fisher Scientific Inc., Waltham, MA, USA) was employed to analyze the species and elemental composition of LIG. The crystallographic characteristics, including the location and intensity of LIG crystal peaks, were determined using a Rigaku Ultima IV X-ray diffraction system (XRD, Rigaku Corporation, Fuji, Japan). The spatial distribution and intensity of the Raman peaks were analyzed using a Renishaw inVia Focal Microscope Raman Spectrometer (Raman Systems Ltd., Gloucestershire, UK). The type and content of LIG were assessed using an FEI Escalab Xi+ X-ray photoelectron spectroscope (XPS, Thermo Fisher Scientific Inc., Waltham, MA, USA). Real-time resistance measurements were performed with a digital multimeter (Rigol Technologies Ltd., Suzhou, China), and strain tests were conducted using a tensile testing machine (WNMC Beijing Co., Ltd., Beijing, China).

## 3. Results and Discussion

### 3.1. Surface Morphology of LIG

The surface morphology analysis of Ag_-x_/LIG, generated using different concentrations of AgNO_3_ solution, was conducted. Figure 2 presents the SEM image of Ag_-x_/LIG, facilitating microscopic observation of the LIG surface. In Figure 2a, both a sample diagram and an SEM image were obtained from untreated PI, serving as a control. SEM magnification revealed that the surface of the LIG was smooth with no visible grains. Figure 2b displays the samples and SEM images of Ag_-0.01_/LIG, revealing small particle distributions on the surface caused by the laser radiation energy impacting the PI surface, and the silver was analyzed at high temperature(>300 °C). Figure 2c shows the Ag_-0.03_/LIG sample along with its SEM image, demonstrating a slightly denser distribution of silver nanoparticles compared to those on the Ag_-0.01_/LIG surface. This increased density may be attributed to the higher concentration of AgNO_3_, facilitating more extensive ion exchange processes, thereby promoting LIG formation at active sites during the laser action experiments at elevated temperatures and resulting in larger nano-silver particle formation.

However, Figure 2d illustrates the SEM image of Ag_-0.05_/LIG, revealing that the elemental silver particles on its surface were less conspicuous than those depicted in Figure 2c. This discrepancy can be attributed to the more complete reaction of PI induced by Ag^+^ during laser experimentation, resulting in an increased generation of LIG. The LIG possesses an abundant network of 3D pores, which potentially contribute to the dispersion of element silver particles and subsequently reduce their visibility on the surface region of the LIG. Thus, Ag_-0.03_/LIG may exhibit superior quality in this experimental setup.

Therefore, SEM and EDS were employed to test the Ag_-0.03_/LIG samples, and the corresponding results are presented in Figure 3. The SEM image depicted in Figure 3a revealed the uniform distribution state of the LIG surface, while the elemental spectrum analysis from Figure 3b–f also confirmed the presence of nanoparticles within the LIG.

### 3.2. Characterization of LIG

Raman, XRD, and XPS tests were conducted for Ag_-0.00_/LIG, Ag_-0.01_/LIG, Ag_-0.03_/LIG, and Ag_-0.05_/LIG samples. The results are depicted in Figure 4. Figure 4a presents the Raman analysis of LIG at varying concentrations, showing the presence of the D peak at 1333.62 cm^−1^, G peak at 1576.53 cm^−1^, and 2D peak at 2678.79 cm^−1^. The higher intensity of the D peak compared to the G peak indicates abundant defects [26], which are further enhanced by silver doping in LIG. The intensities of these peaks were analyzed to obtain the I_D_/I_G_ ratio depicted in Figure 4b. A higher I_D_/I_G_ ratio indicates more defects in the LIG due to the addition of nano-silver from the increasing concentrations of the AgNO_3_ solution used during the synthesis process. The I_2D_ and I_G_ values both exceed unity, suggesting that LIG is multilayered. Figure 4c displays the XRD analysis of LIG, used to analyze the crystal structure. The Ag-ion-modified LIG exhibited a prominent (002) peak at 2θ = 24.98°, corresponding to the characteristic peak of graphene [27], and a weak (100) peak at 2θ = 44.02°, which corresponds to the arrangement of aromatic rings in the carbon material. These distinct peaks confirm the presence of graphene. Additionally, Figure 4d reveals the crystal faces (111), (200), (220), and (311) at angles of 38.04°, 47.16°, 64.4°, and 77.84°, respectively [28], indicating a high purity elemental silver content in the LIG.

XPS analysis was also performed for all concentrations to determine the element species and content of LIG, as shown in Figure 4e. The results exhibited the presence of C, N, and O elements for all samples. Compared to Ag_-0.00_/LIG, the XPS spectra of Ag_-0.01_/LIG, Ag_-0.03_/LIG, and Ag_-0.05_/LIG displayed additional peaks corresponding to Ag3d and Ag3p. The atomic percentage composition of C, N, O, and Ag in LIG at different concentrations is presented in Figure 4f. Notably, Ag_-0.00_/LIG contained no silver elements, whereas Ag_-0.01_/LIG, Ag_-0.03_/LIG, and Ag_-0.05_/LIG all contained Ag. The C1s peak of Ag_-0.03_/LIG can be divided into four sub-peaks: C=C, C-N, C-O, and C=O, as shown in Figure 4g [29]. Similarly, the Ag3d peak was divided into two peaks: Ag3d_3/2_ and Ag3d_5/2_, as demonstrated in Figure 4h. The flexible wearable strain sensor based on Ag_-x_/LIG underwent initial resistance tests with different concentrations of AgNO_3_. As depicted in Figure 4i, a decreasing trend in resistance was observed as the concentration of AgNO_3_ increased, with recorded values of 432.4 Ω, 363.3 Ω, 183.4 Ω, and 107.4 Ω, respectively. These findings suggest that increasing the concentration of AgNO_3_ solution enhances the reduction in the resistance value of the sensor. The significant disparities in error bars indicate that the introduction of varying concentrations of silver exerts a noticeable impact on the resistance of LIG.

### 3.3. Performance Testing of Doping Ag-LIG Sensor

The sensitivity of the flexible wearable strain sensor reflects its ability to detect minute changes in the external environment, specifically its response to applied strain, which is a critical parameter for assessing the sensor’s performance and directly impacts its effectiveness and accuracy in practical applications. The Gauge factor (GF) of the sensor, which indicates sensitivity, is calculated based on the relative resistance change and tensile strain:(1)$$GF=\frac{\delta (R-R_{0})/R_{0}}{\delta \varepsilon }$$
where GF represents the sensitivity factor of the sensor, R is the real-time resistance of the sensor, R_0_ is the initial resistance, (R − R_0_)/R_0_ is the relative resistance change in the sensor, and δε is the tensile strain of the sensor.

Performance testing and evaluation of flexible wearable strain sensors, based on Ag_-x_/LIG prepared with varying concentrations of AgNO_3_ following PI pretreatment, were conducted as shown in Figure 5. Figure 5a explores the strain–relative resistance response characteristics of four distinct samples. Figure 5b focuses on the relative resistance changes in the Ag_-0.03_/LIG–based flexible wearable strain sensor, employing a three–stage function for fitting. The sensitivity factor GF of the sensor was found to be 37.2 within the 0–5% tensile strain range in the first section. In the second section, a significant increase in the sensitivity factor GF to 177.3 was observed within the 5.0–8.0% range. Moreover, as the tensile strain further increased to 8.0–11%, an even higher sensitivity factor GF value of 426.8 was achieved. Figure 5c compares the performance of four types of flexible wearable strain sensors, revealing that Ag_-0.03_/LIG–based sensors exhibit superior GF values. Additionally, as illustrated in Figure 5d, our sensor demonstrated exceptional sensitivity compared to other materials or processes utilized for the preparation of flexible wearable strain sensors [17,30,31,32,33,34,35,36,37,38].

In Figure 5e, the stability test results of the flexible wearable strain sensor under various tensile strains are presented. The experimental results demonstrate that the sensor has an impressive detection limit, as low as 0.2%. Moreover, it has been subjected to ten cycles of testing across a range of tensile strains from 1.0% to 8.0%, at intervals of 1.0%, showcasing exceptional stability. Figure 5f illustrates the time response characteristics of the flexible wearable strain sensor, revealing a response time of approximately 150 ms and a relaxation time of around 200 ms when subjected to a tensile strain of 4.0% for a duration of 5 s at a speed of 500 mm/s, before being released at the same speed. Additionally, cyclic repeatability testing was conducted on the sensor, demonstrating excellent repeatability over an extended period lasting up to 8000 s. These comprehensive test results highlight the sensor’s rapid response and high stability, making it highly suitable for applications that require utmost reliability and prompt feedback.

### 3.4. Application of Ag-LIG Sensor

The flexible wearable strain sensor offers a broad range of applications across various sectors. Specifically, the relative resistance waveform of the sensor demonstrates corresponding patterns. To further explore the variation in relative resistance and the maximum testing angle of the flexible wearable strain sensor under different degrees of bending, this study conducted an analysis, with the findings presented in Figure 6. The initial step involved attaching the sensor to the inner side of a hinge to simulate finger flexion at various bending angles. As shown in Figure 6a, when the hinge bends, the sensor also bends outwardly along with its protruding LIG surface, leading to changes in the conductive path and subsequent alterations in relative resistance. Bending was performed at 30° intervals, starting from 0° and maintained at each angle for approximately 15 s up to a maximum bend angle of 120°. Subsequently, the bend was gradually released at equal intervals. The relative changes in resistance were measured as depicted in Figure 6b, demonstrating that similar variations occur within identical bending angles for this device. The results underscore the sensor’s effective measurement capability across various levels of finger flexion, while maintaining excellent stability and reversibility characteristics. Figure 6c illustrates θ as a reliable parameter for quantifying finger flexion angles, and Figure 6d depicts the relative resistance changes associated with utilizing this device to assess such angles. Notably, there is an incremental increase in resistance with every additional 15° increment in bending. The change in relative resistance of the flexible wearable strain sensor is depicted in Figure 6e, as it is subjected to force while being attached to the back of the hand. The gripometer encompasses seven weight categories, ranging from a minimum of 5 kg to a maximum of 60 kg. Notably, an increase in the gripometer’s gravity scale corresponds to an elevation in the relative resistance exhibited by the sensor. Therefore, it can be concluded that this versatile and adaptable technology provides valuable insights into accurately measuring finger flexion angles.

The infrared thermal imager was employed for real-time monitoring of the Ag_-0.03_/LIG-based flexible wearable strain sensor to evaluate temperature variations and electric heating performance, as illustrated in Figure 7. The graph in Figure 7a demonstrates a continuous increase in maximum surface temperature with incremental voltage increases, reaching a peak of 119.2 °C at a setting of 10 V. Figure 7b provides a direct assessment of the temporal and voltage-dependent changes in surface temperature, clearly showing an increasing maximum temperature as the voltage rises. Simultaneously, different voltages induce a rapid initial temperature rise within the first 10 s, followed by significantly reduced fluctuations that converge toward stability thereafter. By examining Figure 7c, it can be observed that at a voltage of 2 V, the surface temperature of the sensor approximated the ambient temperature (28.8 °C). As the voltage gradually increased, there was a corresponding increase in the maximum surface temperature until it reached its final value at 10 V (146.4 °C). Figure 7d demonstrates an upward correlation between the surface temperature of the Ag_-0.03_/LIG-based flexible wearable strain sensor and the applied voltage. Furthermore, when comparing flexible wearable strain sensors based on Ag_-0.03_/LIG and LIG, it becomes apparent that nano-silver enhances conductivity and reduces heating time due to its excellent electrical conductivity properties. The enhanced heat generation can be attributed to the infiltration of nano-silver into the three-dimensional nanopore structure of LIG, effectively reducing overall resistance and enhancing current conduction efficiency within the material.

The excellent electric heating performance of the flexible wearable strain sensor enables its extensive application. It can be utilized not only for pilot flight training in indoor simulators at normal temperatures or for monitoring other human physiological signals, but also for hand warming and maintenance in cold environments. In extremely low temperatures, pilots may experience adverse effects on their manual dexterity and response time due to compromised finger flexibility, potentially compromising flight safety. The integration of flexible wearable strain sensors into gloves or hand-worn devices, along with the utilization of their electric heating function, enables the provision of continuous and comfortable hand warming to pilots during flights. The implementation of this solution effectively alleviates the discomfort caused by cold conditions on pilots’ hands, thereby enhancing hand comfort and reaction speed. Ultimately, this contributes significantly to ensuring flight safety.

## 4. Conclusions

This study aimed to enhance the characteristics of LIG through Ag doping modification. The PI substrate was pretreated with AgNO_3_ solutions at concentrations of 0.01, 0.03, and 0.05 mol/L to introduce Ag ions onto its surface, followed by laser experiments. The SEM, XRD, and XPS analyses revealed the presence of uniformly distributed silver nanoparticles on the LIG surface. Tensile performance testing demonstrated that the Ag_-0.03_/LIG-based flexible sensor exhibited the highest sensitivity, reaching 426.8, representing a significant improvement compared to the undoped LIG-based flexible sensor. The subsequent performance tests confirmed the sensor’s outstanding stability and repeatability, suggesting its potential application in specific fields for experimental verification and practical use. This experiment provides a novel approach for the in situ preparation of LIG and the fabrication of high-performance sensors in a single step.

## Figures and Tables

**Figure 1 micromachines-15-00989-f001:**
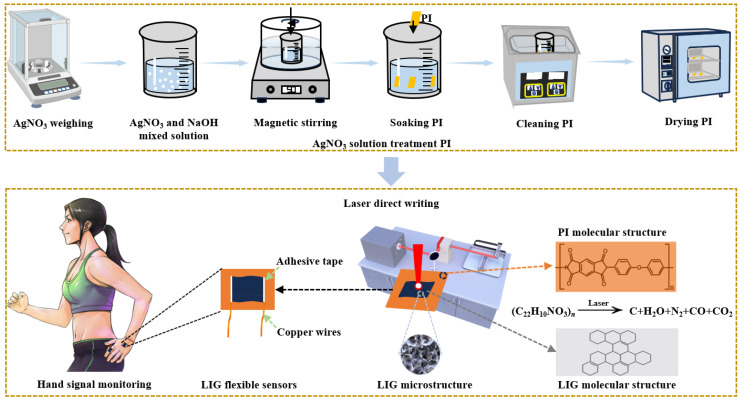
Experimental process of LIG.

**Figure 2 micromachines-15-00989-f002:**
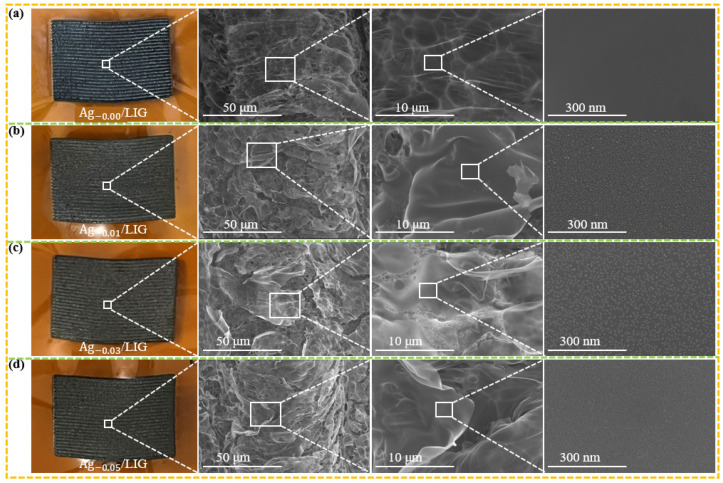
SEM detection of Ag_-x_/LIG. (**a**) Ag_-0.00_/LIG, (**b**) Ag_-0.01_/LIG, (**c**) Ag_-0.03_/LIG, (**d**) Ag_-0.05_/LIG.

**Figure 3 micromachines-15-00989-f003:**
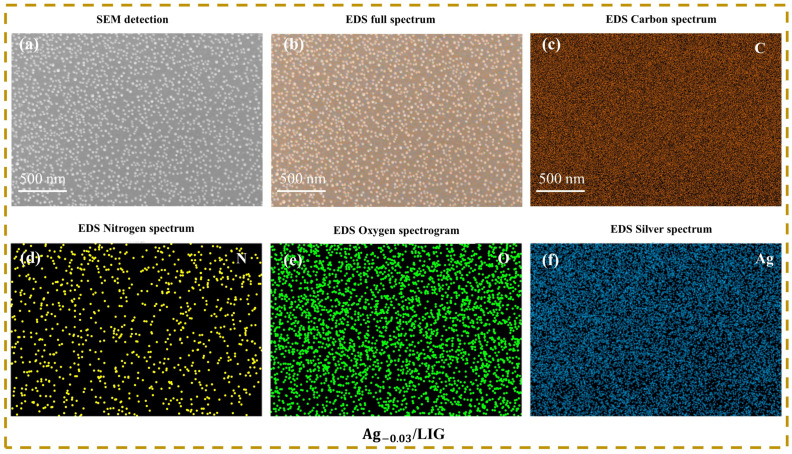
SEM and EDS detection of Ag_-0.03_/LIG. (**a**) SEM detection, (**b**–**f**) full spectrum and partial spectrum of EDS.

**Figure 4 micromachines-15-00989-f004:**
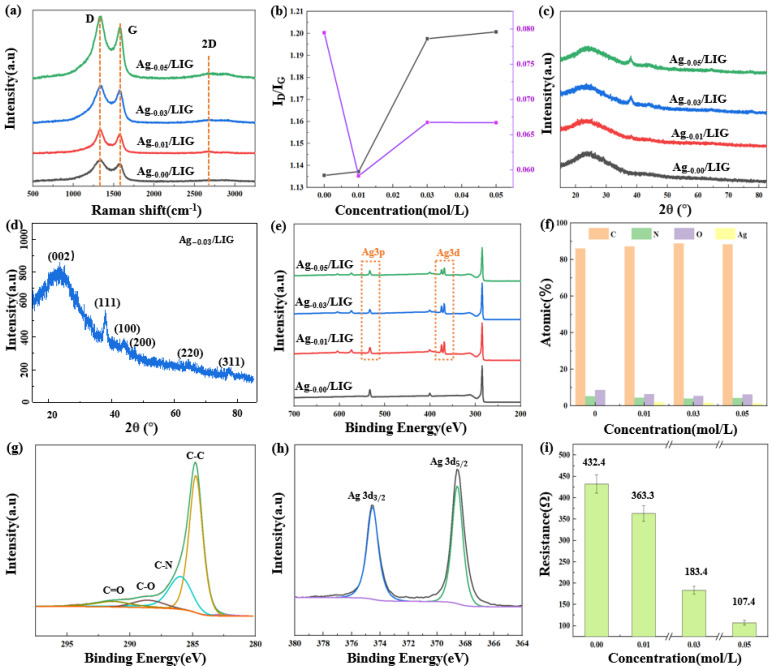
Characterization of Ag_-x_/LIG. (**a**) Raman spectra, (**b**) calculation value of I_D_/I_G_ and I_2D_/I_G_, (**c**) XRD detection, (**d**) detection of Ag_-0.03_/LIG, (**e**) XPS detection, (**f**) element contnt percentage, (**g**) C1s sub–peak and fitting, (**h**) Ag3d sub–peak and fitting, (**i**) resistance of Ag_-x_/LIG.

**Figure 5 micromachines-15-00989-f005:**
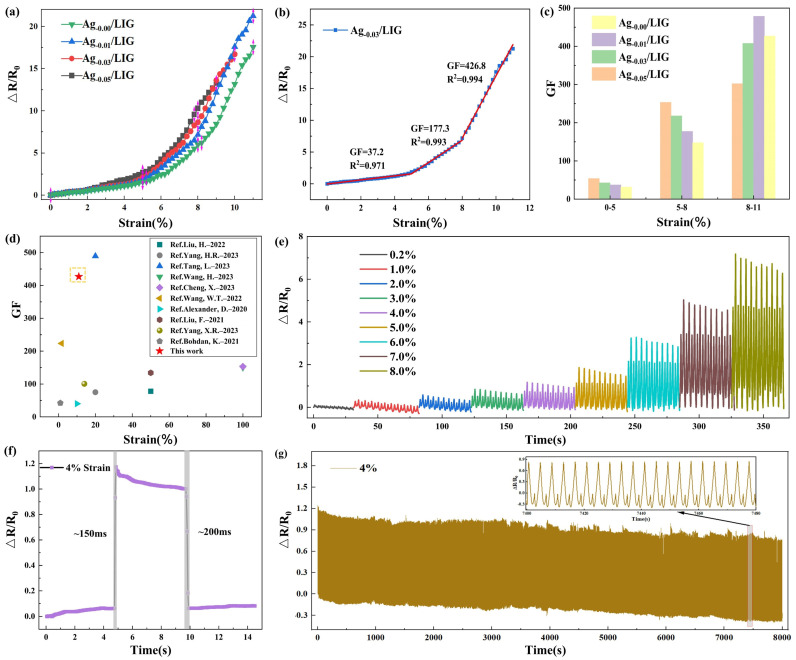
Tensile performance tests of the flexible wearable strain sensor. (**a**) Tensile strain and reltive resistance tests, (**b**) tensile strain and relative resistance change test of Ag_-0.03_/LIG–based flexible wearable strain sensor, (**c**) sensitivity comparison of four flexible wearable strain sensors, (**d**) senstivity comparison of Ag_-0.03_/LIG–based flexible wearable strain sensors with other materials or processes [17,31,32,33,34,35,36,37,38], (**e**) stability testing at different tensile lengths, (**f**) time response testing, (**g**) cyclic repeatability testing.

**Figure 6 micromachines-15-00989-f006:**
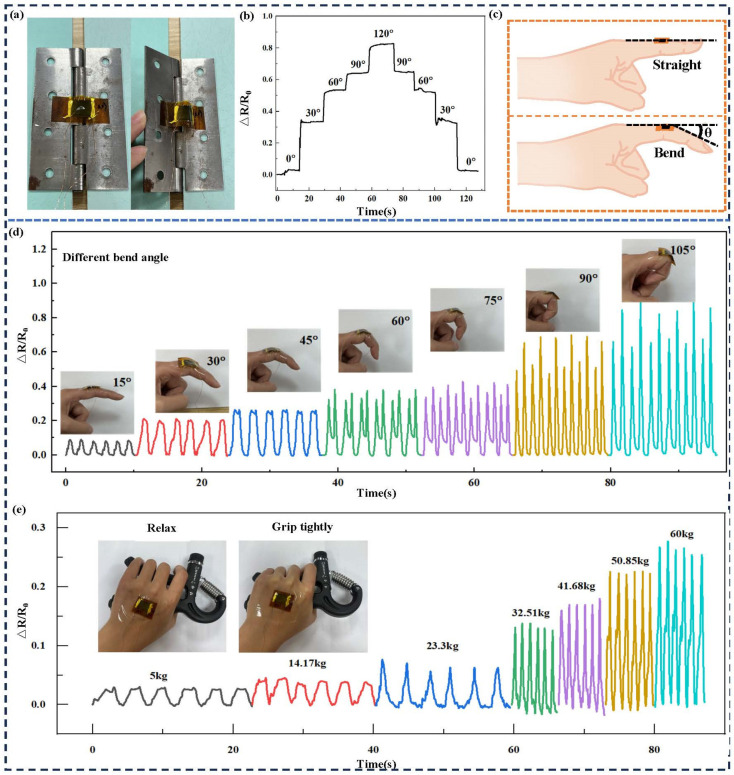
Changes in relative resistance of flexible wearable strain sensors under different tensile forces. (**a**) Testing the bending condition of the flexible strain sensor in the middle of the hinge, (**b**) the relative resistance change during the bending process of the flexible strain sensor in the hinge, (**c**) schematic diagram of finger bending angle, (**d**) testing the relative resistance changes in flexible strain sensors during finger bending at different angles, (**e**) the relative resistance change in the flexible strain sensor during the use of the grip strength meter in the test hand.

**Figure 7 micromachines-15-00989-f007:**
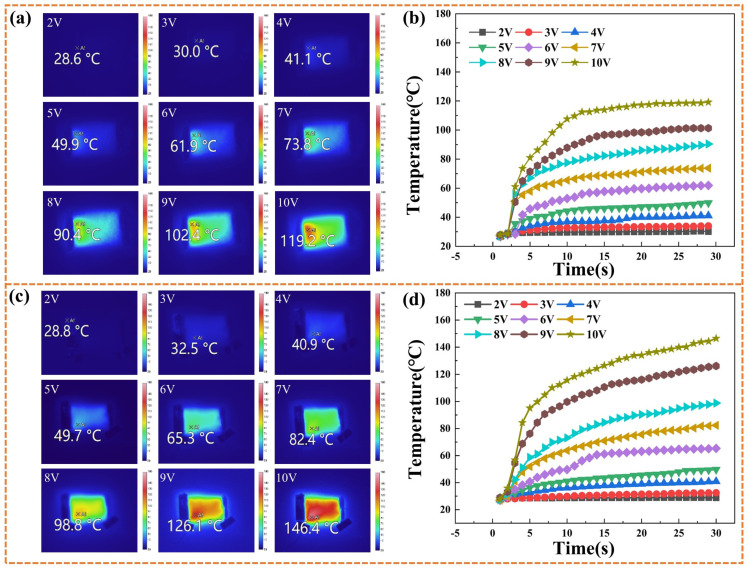
Thermal performance test and temperature change curve of LIG-based and Ag_-0.03_/LIG–based flexible wearable strain sensor. (**a**) Infrared thermal performance test of LIG, (**b**) temperature change curve during the electric heating performance test of LIG, (**c**) infrared thermal performance test of Ag_-0.03_/LIG, (**d**) temperature-change curve during the electric heating performance test of Ag_-0.03_/LIG.

## Data Availability

The data supporting this study’s findings are available from the corresponding author upon reasonable request.

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
