# Peer review of "A Flexible Wearable Strain Sensor Based on Nano-Silver-Modified Laser-Induced Graphene for Monitoring Hand Movements"

_micromachines, 2024, doi:10.3390/mi15080989_

Round 1

Reviewer 1 Report

Comments and Suggestions for Authors

A laser-induced graphene-based pressure sensor is proposed by the authors. The explicit design and fabrication strategies are provided, and practical pressure/force measurements are presented. I would make my recommendation after the following concerns are well addressed:

1,I think the Introduction of this paper should be more specific, especially regarding the research progress of LIG.

2,Page 2, 64. MoS2 should be MoS2.

3,Page 3, 130-132. The abundant three-dimensional pore structure of LIG disperses elemental silver particles, reducing their visibility on the surface area of LIG. Can the author experimentally prove the reliability of the viewpoint?

4,The process and parameters of laser processing in LIG preparation should be simply supplemented.

5,The error bars in Figure 4i are not explained.

6,How is tensile strain in Figure5a obtained? Can tensile and compressive testing machines directly obtain micro strain? Or can strain be converted by detecting force? I hope to have a simple understanding of this process.

7,Figure6e. Does the sensor response come from grip strength or hand clenching movements? Or is it a combination of the two? How does the sensor respond if you don't hold the gripometer, only the hand grip? The decoupling of grip strength and strain can be properly considered.

8,Figure7. Has the author paid attention to the resistance response of sensors during the electric heating process? Although the electric heating performance of the sensor has potential for development in pilot driving, this characteristic should be detected in electrical signals to be meaningful.

Comments on the Quality of English Language

Amelioration on writing is suggested.

Author Response

Response to Reviewer 1 (Micromachines-3106480)

A laser-induced graphene-based pressure sensor is proposed by the authors. The explicit design and fabrication strategies are provided, and practical pressure/force measurements are presented. I would make my recommendation after the following concerns are well addressed:

  1. I think the Introduction of this paper should be more specific, especially regarding the research progress of LIG.

The authors’ answer 1: Thank you very much for your valuable suggestion. We have incorporated a dedicated section in the introduction of the revised manuscript that offers an extensive overview of significant discoveries and advancements in the field of LIG, thereby enhancing comprehension regarding the background and significance of our research. (Page 2, lines 65-77, highlighted in yellow)

  1. Page 2, 64. MoS2 should be MoS2.

The authors’ answer 2: Thanks for your careful advice. The text has been revised. (Page 2, lines 78, highlighted in yellow)

  1. Page 3, 130-132. The abundant three-dimensional pore structure of LIG disperses elemental silver particles, reducing their visibility on the surface area of LIG. Can the author experimentally prove the reliability of the viewpoint?

The authors’ answer 3: Thanks for your constructive suggestions. By comparing the SEM images of Ag-0.03/LIG and Ag-0.05/LIG in the paper, it can be observed that the surface of Ag-0.05/LIG exhibits a reduced presence of visible silver particles. The SEM image below demonstrates that LIG possesses an abundant network of three-dimensional pore structure under these experimental conditions. Therefore, we hypothesize that the silver particles are effectively dispersed within the three-dimensional porous structure of LIG, thereby reducing their visibility on the surface. However, we regret to inform you that we are currently unable to further validate this hypothesis due to certain limitations: (1) Our relocation to a new campus from July to August has resulted in equipment scattering and unavailability as it is yet to be calibrate; (2) Our partner institutions are currently on summer break, preventing us from conducting experiments at research platforms. We plan to carry out experimental validation in September once the equipment is fully operational again. Please rest assured that we adhere strictly to rigorous scientific standards and ensuring authenticity remains our ultimate goal in research. (Page4, lines 162-165, highlighted in yellow)

  1. The process and parameters of laser processing in LIG preparation should be simply supplemented.

The authors’ answer 4: The revised version of the manuscript now includes a detailed description of the pertinent laser parameters. (Page 3, lines 116-124, highlighted in yellow)

  1. The error bars in Figure 4i are not explained.

The author’s answer 5: Thank you for your valuable advice. By incorporating error bars, we can effectively compare the resistance changes associated with different silver contents. Figure 4(i) clearly demonstrates significant variation in the resistance of LIG treated with various concentrations of AgNO3, as evidenced by noticeable disparities in the error bars. This observation strongly indicates that silver plays a crucial role in reducing resistance. The detailed descriptions have been incorporated into the relevant section of the revised manuscript. (Page 7, lines 216-217, highlighted in yellow)

  1. How is tensile strain in Figure5a obtained? Can tensile and compressive testing machines directly obtain micro strain? Or can strain be converted by detecting force? I hope to have a simple understanding of this process.

The authors’ answer 6: Utilizing a universal testing machine, we can directly stretch a flexible strain transducer to a specified length and subsequently release it. The relationship between the strain and the stretched length is described by the following formula.

where L represents the length of the flexible wearable strain sensor after stretching, L0 is the original length of the sensor, and ε denotes the strain. This formula enables us to calculate the strain by utilizing the alteration in length of the sensor.

  1. Figure Does the sensor response come from grip strength or hand clenching movements? Or is it a combination of the two? How does the sensor respond if you don't hold the gripometer, only the hand grip? The decoupling of grip strength and strain can be properly considered.

The authors’ answer 7: After calibrating the dynamometer to a specific weight, when firmly grasping the dynamometer handle and subsequently releasing it, the sensor material (LIG) on the dorsal surface of the hand will detect subtle changes in skin topography. If only gripping the handle without applying force to the dynamometer, there will also be an increase in sensor readings. Additionally, even during a stationary grip on the handle, slight variations in sensor readings will still be observed compared to pre-grip measurements.

  1. Has the author paid attention to the resistance response of sensors during the electric heating process? Although the electric heating performance of the sensor has potential for development in pilot driving, this characteristic should be detected in electrical signals to be meaningful.

The authors’ answer 8: Thank you for your valuable suggestions. Firstly, the electric heating test section solely focus on voltage, temperature, and time without mentioning resistance.When the resistance of LIG decreases, indicating an increase in conductivity, it enhances the efficiency of current passing through LIG material. According to Joule's law, heat is generated as current flows through a conductor and its magnitude is directly proportional to both the square of the current and the resistance. Consequently, when resistance decreases and current increases, there is a corresponding increase in Joule heat generation leading to elevated temperature on the surface of LIG. Furthermore, incorporating silver nanoparticles also exerts a certain influence. The introduction of doped silver nanoparticles significantly enhances thermal and electrical conductivity in LIG materials, thereby increasing their temperature rise rate under identical voltage conditions. By leveraging silver’s high thermal conductivity properties, doped LIG exhibits excellent heat transfer capabilities throughout its entirety upon application of voltage.

Additionally, the strain and relative resistance variations observed in flexible strain sensors play a crucial role in pilot monitoring and motion capture application, similarly, the electric heating functionality holds great promise for such devices as well. Superior electric heating performance may find potential application in hand warming and insulation fields alike. We are intended to validate these findings during subsequent stages.

Reviewer 2 Report

Comments and Suggestions for Authors

(1)      In Section 2 on Page 2, there is no description of basic characterization methods, including Raman, XRD, XPS, resistance and performance measurements which the authors didn’t mention too much in the following discussion.

(2)     In Line 103 on Page 3, what is the basic laser parameters for the experiments? such as laser type, frequency, spot size and power, etc. because it’s important to prepare both LIG and nano-Ag.

(3)     In Figure 3e and 3f on Page 4, the images are too dark, and increase the brightness of blue and green.

(4)     In Line 147 on Page 4 and Figure 4a on Page 5, the authors need clarify and compare the Raman shift of LIG and nano-Ag, explain why the 2D peak is so weak and the D peak is higher than G peak for LIG, because the Raman shift of Ag is almost in the same positions with LIG, check the overlap and difference of Raman peaks of Ag_0.00/LIG and Ag_0.01/LIG.

(5)     In Figure 4d on Page 5, the position of the (100) peak is less than 40°, not exactly at 2θ = 44.02°, please verify it.

(6)     In Figure 5e, 5f and 5g on Page 6, what samples were used in these experiments, especially for 4% strain? Because the GF of Ag_0.03/LIG in 4% strain is not the highest.

(7)     In Figure 6 on Page 7, each picture should be described in the title of figure.

(8)     In Figure 7b and 7d on Page 8, the unit of time in the X axis needs label.

(9)     In Line 276 on Page 9, is the overall resistance mentioned here different from the initial resistance in Figure 4i? the explanation is not clear, please analyze why the heating temperature of lower resistant Ag-0.03/LIG is higher than that of LIG.

Comments on the Quality of English Language

The manuscript entitled “A Flexible Wearable Strain Sensor Based on Nano-Silver Modified Laser-Induced Graphene for monitoring Hand Movements” is interesting and carefully written, the experimental data, results and discussion in the manuscript are carefully analyzed and basically reasonable, the paragraph and expression are well organized.

Author Response

Response to Reviewer 2(Micromachines-3106480)

  1. In Section 2 on Page 2, there is no description of basic characterization methods, including Raman, XRD, XPS, resistance and performance measurements which the authors didn’t mention too much in the following discussion.

The authors’ answer 1: Thank you very much for your constructive suggestions. We have added the relevant information to the relative sections of the revised manuscript. (Page 4, lines 130-145, highlighted in green)

  1. In Line 103 on Page 3, what is the basic laser parameters for the experiments? such as laser type, frequency, spot size and power, etc. because it’s important to prepare both LIG and nano-Ag.

The author’s answer 2: Thank you very much for your valuable suggestions. We have incorporated the relevant information regarding laser parameters into the corresponding section of the revised manuscript. (Page 3, lines 121-124, highlighted in green)

  1. In Figure 3e and 3f on Page 4, the images are too dark, and increase the brightness of blue and green.

The author’s answer 3:Thank you for your careful advice. We have adjusted the brightness and contrast of the images, as shown in Figure 3. Additionally, we have replaced the images in the corresponding sections of the revised manuscript.

(Page5, Figure 3)

Original Figure 3                      Revised Figure 3

  1. In Line 147 on Page 4 and Figure 4a on Page 5, the authors need clarify and compare the Raman shift of LIG and nano-Ag, explain why the 2D peak is so weak and the D peak is higher than G peak for LIG, because the Raman shift of Ag is almost in the same positions with LIG, check the overlap and difference of Raman peaks of Ag_0.00/LIG and Ag_0.01/LIG.

The authors’ answer 4: Thank you for your valuable suggestions.

(1) The Raman spectrum in the image shows a higher intensity of the D peak compared to the G peak. This observation can be attributed to a slight insufficient in laser irradiation energy during the addition of Ag to Ag-0.03/LIG and Ag-0.05/LIG, resulting in an increased presence of defects within pure LIG. The introduction of Ag further contributes to an elevated number of defects in LIG due to its presence.

(2) Regarding the Raman shift, we amplified the peak positions corresponding to the characteristic peaks of silver for confirmation but did not detect any Ag peaks. However, EDS analysis in Figure 3 confirms the presence of Ag on the LIG surface. We speculate that the absence of detectable Ag in the Raman spectrum might be attributed to performing Raman testing on a powdered sample, while silver was primarily distributed on the LIG surface and present in low quantities. When pulverized, Ag may have been dispersed, and it is possible that only a fraction was detected, potentially missing its presence. (Page 5, lines 182-186, highlighted in green)

  1. In Figure 4d on Page 5, the position of the (100) peak is less than 40°, not exactly at 2θ = 44.02°, please verify it.

The author’s answer 5: Thank you for your valuable suggestions. Upon thorough examination, we have confirmed that it was indeed marked incorrectly and have promptly rectified the error. (Page 6, Figure 4(d))

Original Figure 4(d)               Revised Figure 4(d) 

  1. In Figure 5e, 5f and 5g on Page 6, what samples were used in these experiments, especially for 4% strain? Because the GF of Ag_0.03/LIG in 4% strain is not the highest.

The authors’ answer 6: Thank you for your valuable suggestions. The stretchability of PI is relatively inferior compared to other flexible materials, whereas the 4% tensile strain exhibits significant higher stability in contrast to strains of 0.2%, 1.0%, 2.0%, and 3.0% as observed in ten stability test results. Utilizing the maximum tensile strain for cyclic stability testing may lead to rapid damage of the flexible sensor. Therefore, it becomes imperative for us to investigate a more resilient flexible substrate.

  1. In Figure 6 on Page 7, each picture should be described in the title of figure.

The authors answer 7: Thank you for your valuable suggestions. We have added a description in the corresponding section of the revised manuscript.

(Page 8, lines 265-270, highlighted in green)

  1. In Figure 7b and 7d on Page 8, the unit of time in the X axis needs label.

The author’s answer 8: Thank you for your considerate suggestion. We have made modification in Figure 7. (Page 9, Figure 7)

Original Figure 7               Revised Figure 7

  1. In Line 276 on Page 9, is the overall resistance mentioned here different from the initial resistance in Figure 4i? the explanation is not clear, please analyze why the heating temperature of lower resistant Ag-0.03/LIG is higher than that of LIG.

The author’s answer 9: Thank you for your valuable suggestions. In the electrothermal testing section, we investigated the correlation between voltage, temperature, and time. According to the formula QJ=Pt= (U2/R)t, as the resistance of LIG decreases, there is an increase in Joule heat generation, resulting in a corresponding rise in the surface temperature of LIG. Additionally, doping with silver nanoparticles can significantly enhance both the thermal and electrical conductivity of the material, thereby accelerating the rate of temperature increase under identical voltage conditions. By leveraging the high thermal conductivity of silver, doped LIG facilitates more efficient heat transfer throughout the material. Importantly, we observed that the sample of Ag-0.05/LIG caused damage due to excessive heating during the electrical heating experiment. Therefore, we opted for Ag-0.03/LIG for further investigation.
